# Highly bright perovskite light-emitting diodes enabled by retarded Auger recombination

Zhiqi Li[1,2,3,8], Qi Wei[4,8], Yu Wang[1], Cong Tao[5], Yatao Zou[5], Xiaowang Liu[5], Ziwei Li[6], Zhongbin Wu[5], Mingjie Li[4] ✉, Wenbin Guo[3] ✉, Gang Li[2] ✉, Weidong Xu[5,7] ✉ & Feng Gao[1] ✉

One of the key advantages of perovskite light-emitting diodes (PeLEDs) is their potential to achieve high performance at much higher current densities compared to conventional solution-processed emitters. However, state-of-the-art PeLEDs have not yet reached this potential, often suffering from severe current-efficiency roll-off under intensive electrical excitations. Here, we demonstrate bright PeLEDs, with a peak radiance of 2409 W sr$^{-1}$ m$^{-2}$ and negligible current-efficiency roll-off, maintaining high external quantum efficiency over 20% even at current densities as high as 2270 mA cm$^{-2}$. This significant improvement is achieved through the incorporation of electron-withdrawing trifluoroacetate anions into three-dimensional perovskite emitters, resulting in retarded Auger recombination due to a decoupled electron-hole wavefunction. Trifluoroacetate anions can additionally alter the crystallization dynamics and inhibit halide migration, facilitating charge injection balance and improving the tolerance of perovskites under high voltages. Our findings shed light on a promising future for perovskite emitters in high-power light-emitting applications, including laser diodes.

Metal halide perovskite emitters are emerging as promising candidates for next-generation light-emitting diodes (LEDs) due to their low cost, high color purity, tunable emission wavelength, compatibility with low-temperature processing, and flexible module fabrication[1–7]. Over recent years, numerous strategies have been developed to improve the external quantum efficiencies (EQE) of perovskite LEDs (PeLEDs), including defect passivation[8], light outcoupling management[9], interface engineering[10], compositional and dimensional tailoring[11,12], as well as optimization of crystallization

processes[13], boosting the EQE to more than 20%[14,15] and gradually approaching state-of-the-art organic LEDs (OLEDs)[16,17]. Beyond achieving high peak quantum efficiencies, it has been generally believed that the perovskite emitters can potentially achieve better performance at high current density compared to OLEDs, making them desirable for high-power applications, such as outdoor display, lighting and potentially laser diodes[18,19].

Despite the fact that the peak brightness of PeLEDs is usually higher than that of OLEDs, their performance remains significantly

[1]Department of Physics, Chemistry and Biology (IFM), Linköping University, Linköping, Sweden. [2]Department of Electrical and Electronic Engineering, Photonic Research Institute (PRI), Research Institute of Smart Energy (RISE), The Hong Kong Polytechnic University, Hong Kong, China. [3]State Key Laboratory of Integrated Optoelectronics, College of Electronic Science and Engineering, Jilin University, Changchun, China. [4]Department of Applied Physics, The Hong Kong Polytechnic University, Hong Kong, China. [5]Frontiers Science Center for Flexible Electronics, Institute of Flexible Electronics (IFE), Northwestern Polytechnical University, Xi'an, China. [6]Hunan Institute of Optoelectronic Integration, College of Materials Science and Engineering, Hunan University, Changsha, China. [7]Henan Institute of Flexible Electronics (HIFE) and School of Flexible Electronics (SoFE), Henan University, 379 Mingli Road, 450046 Zhengzhou, China. [8]These authors contributed equally: Zhiqi Li, Qi Wei. ✉e-mail: ming-jie.li@polyu.edu.hk; guowb@jlu.edu.cn; gang.w.li@polyu.edu.hk; ifewdxu@nwpu.edu.cn; feng.gao@liu.se

below expectations due to a pronounced decrease in quantum efficiencies over increasing current densities[20,21]. This current-efficiency roll-off can be mainly attributed to Auger recombination—the primary nonradiative paths in conventional III-V quantum wells and chalcogenide quantum dot LEDs—also playing a critical role in perovskites[18,22,23]. For typical three-dimensional (3D) perovskites, the Auger recombination constant is $\sim 10^{-27} - 10^{-28}\,\mathrm{cm^6\,s^{-1}}$, markedly faster by two to three orders of magnitude than III-V quantum well emitters ($\sim 10^{-30}\,\mathrm{cm^6\,s^{-1}}$)[24,25]. This effect is even more pronounced in prevailing mixed-dimensional and quantum dot perovskite emitters due to enhanced carrier confinement and localization[26,27]. More seriously, unbalanced charge injection and relevant leakages in working LEDs exacerbates Auger recombination even at low current densities, due to enhancement in many-body interactions[28]. In addition, fast degradation of PeLEDs at high driving voltages is another critical issue limiting their performance at high current densities as a result of facilitated ion movement and increased Joule heating[29-32]. As such, to achieve high performance of PeLEDs under high current densities, so that high-power applications and electrically pumped lasers can be realized, it is necessary to simultaneously mitigate Auger recombination and device degradation at high excitations.

In this report, we demonstrate bright and highly stable PeLEDs by simultaneously slowing down Auger recombination, balancing charge injection and suppressing ion migration. Specifically, we have managed to decrease the Auger recombination constant by an order of magnitude in 3D perovskite emitters, achieved by the reduced exciton

binding energies in perovskites and the alleviation of charge accumulation at interfaces. Additionally, ion migration is effectively suppressed by introducing additional ionic interactions. All these effects are realized by incorporating trifluoroacetate anions in well-passivated 3D perovskites with dense thin-film coverage. Consequently, the champion PeLED shows a negligible current-efficiency roll-off up to $2000\,\mathrm{mA\,cm^{-2}}$, yielding a high radiance of $2409\,\mathrm{W\,sr^{-1}\,m^{-2}}$. The high radiance is also coupled with excellent operational stability with a half-lifetime of 142 h under a large current density of $100\,\mathrm{mA\,cm^{-2}}$, representing highly bright and highly stable direct-current (DC) driven PeLEDs.

## Results
### Device performance
We use formamidinium lead tri-iodide (FAPbI$_3$) as the parental perovskites, with 5-ammonium valeric acid iodide (5AVAI)[33,34] and tri-fluoroacetate cesium (CsTFA)[35,36] as the additives to enhance the film qualities. The perovskite films were prepared by spin-coating a mixture of FAI, PbI$_2$, 5AVAI, CsTFA with a ratio of 1.28:1:0.1:0.12 in a DMF:DMSO solution (details in methods). The addition of 5AVAI and an extra amount of FAI is crucial for the formation of the α-phase of FAPbI$_3$[33,34]. Unless otherwise stated, we refer to the samples without and with CsTFA as F- and FCT-films/devices, respectively. We also compared the samples enriched with CsI in the same molar ratio as CsTFA (0.12 equivalent to Pb$^{2+}$) to assess the contribution of the TFA$^-$ anions to the film properties (Supplementary Fig. 1).

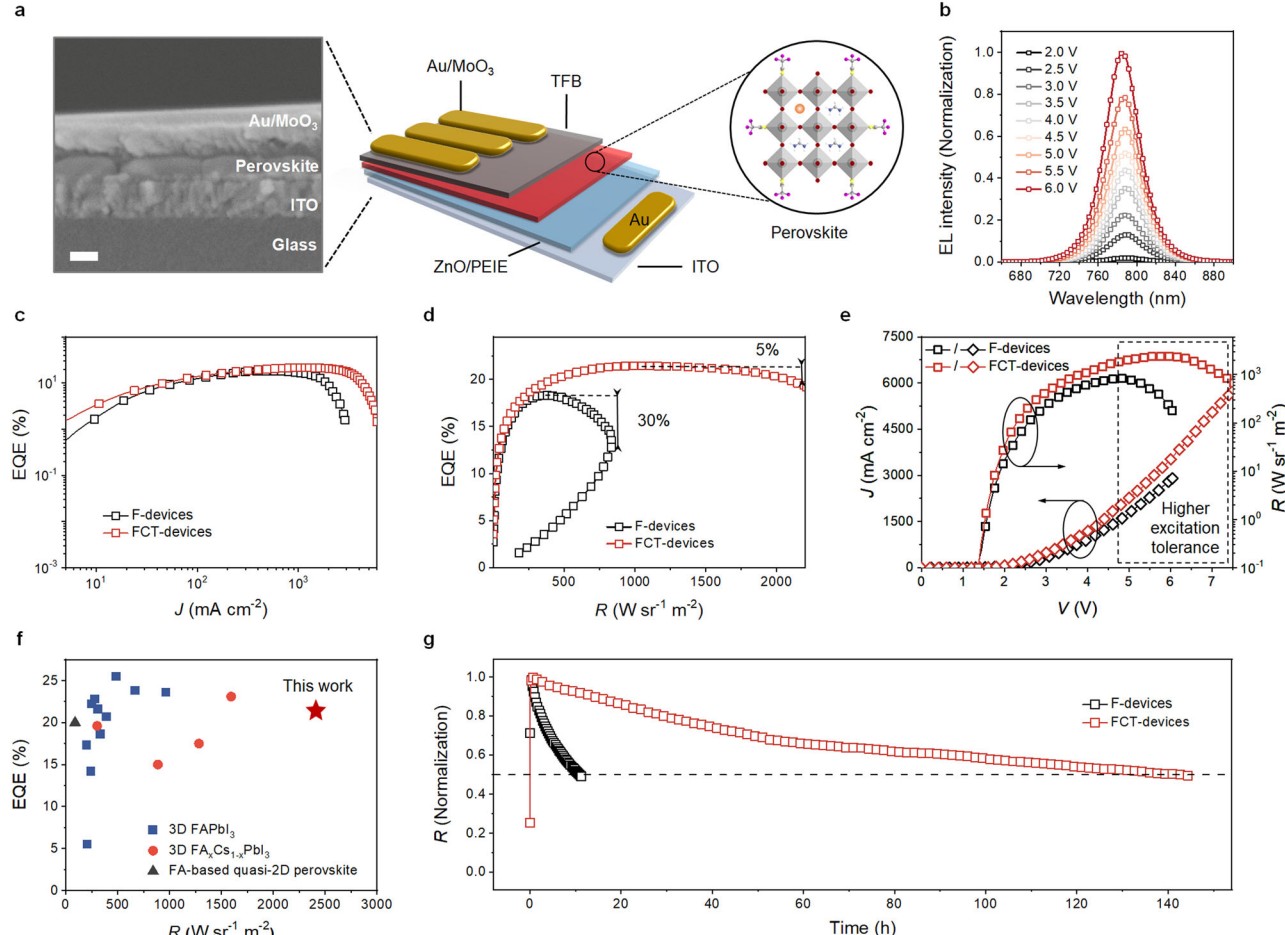

**Fig. 1 | Device structure and performance characteristics of our PeLEDs.**
**a** Device configuration and cross-sectional SEM image of the PeLED. The scale bar represents 100 nm. **b** Electroluminescence (EL) spectra of the device at different voltages. **c** Dependence of EQE on the current density (EQE–J). **d** Dependence of EQE on the radiance (EQE-R). **e** Dependence of current density and radiance on the voltage (J–V–R). **f** A comparison of device efficiency and radiance with the state-of-the-art NIR PeLEDs reported in the literatures (Supplementary Table 1). **g** Stability of the device measured at a high current density of $100\,\mathrm{mA\,cm^{-2}}$.

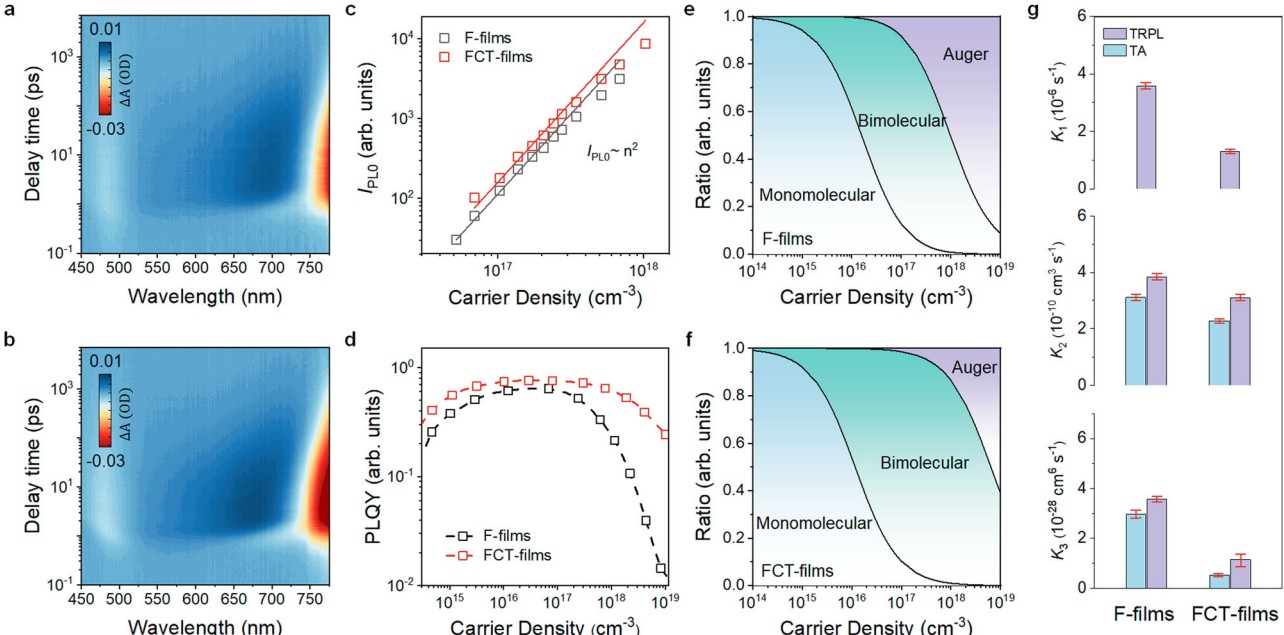

**Fig. 2 | Carrier recombination dynamics of perovskite films.** TA spectra for (**a**) F- and (**b**) FCT-films, respectively. **c** Power dependent $I_{PLO}$ for F- and FCT-films. The $I_{PLO}$ values were extracted from TRPL spectra at time zero. **d** PLQY as a function of excitation power density for F-and FCT-films. Proportion of the recombination ratio as a function of carrier density for the F- (**e**) and FCT-films (**f**), respectively. **g** Derived values of recombination rate constant for the F- and FCT-films, where $k_1$ was extracted from low-fluence TRPL spectra, and $k_2$ and $k_3$ were extracted from both TRPL and TA measurements. The error bars indicate the confidence intervals of the fitted rates.

PeLEDs were fabricated with the multilayered device structure, consisting of indium tin oxide (ITO)/polyethylenimine ethoxylated (PEIE)-modified zinc oxide (ZnO)/perovskite/poly(9,9-dioctyl-fluor-ene-co-N-(4-butylphenyl) diphenylamine) (TFB)/molybdenum oxide (MoO$_3$)/Au (Fig. 1a). The cross-sectional scanning electron microscope (SEM) image (Fig. 1a) shows the formation of dense perovskites films in the emitting layer with a thickness of 80 nm. Our PeLEDs have been engineered to ensure exceptional electrical excitation tolerance and maintain stable electroluminescence (EL) spectra even under high driving voltages of up to 6.0 V, as shown in the characteristics of EL and EQE versus voltage curves (Fig. 1b and Supplementary Fig. 2, respectively). The FCT-devices show a notably higher peak EQE of 21.4% than that of F-ones (~18.9%). With increasing current density, the EQE values of FCT-devices remain above 20% until reaching a current density as high as 2270 mA cm$^{-2}$ (Fig. 1c). The devices exhibit minimal current-efficiency roll-off with EQE reductions of 5% at a high radiance of around 2000 W sr$^{-1}$ m$^{-2}$, a stark contrast to the significant drop in control devices (Fig. 1d). The negligible current-efficiency roll-off gives rise to a high peak radiance of 2409 W sr$^{-1}$ m$^{-2}$ at a current density of 3368 mA cm$^{-2}$ (Fig. 1e). We believe that such a decent performance caused by CsTFA addition is mainly achieved by their TFA$^-$ anions instead of Cs$^+$ cations, as the addition of only CsI leads to a mild effect (Supplementary Fig. 1). We compare our device efficiency/radiance with the state-of-the-art NIR PeLEDs, which is shown in Fig. 1f and further summarized in Supplementary Table 1. In short, the best reported high-performance PeLEDs with EQE over 20% commonly show a moderate radiance[2,7,8,37], and our results represent a significant improvement for DC-driven PeLEDs.

We next measured the operational lifetime of F- and FCT-PeLEDs with a high constant current density of 100 mA cm$^{-2}$. State-of-the-art NIR PeLEDs with the best stability, which showed a half operational lifetime ($T_{50}$) of 11,539 h at 5 mA cm$^{-2}$, demonstrated 120.3 h at such a harsh working condition of 100 mA cm$^{-2}$ for accelerating degradation[37]. As shown in Fig. 1g, while the F-devices exhibit a short $T_{50}$ of 11 h, our FCT-counterparts deliver a drastically enhanced

lifetime reaching 142 h. The excellent stability of our optimized devices is also indicated by the evolution of peak EQE upon voltage scanning cycles (0 - 6.0 V). As shown in Supplementary Fig. 3, the peak EQE of F-devices drop significantly after several cycles. In contrast, the peak EQE of FCT-devices can preserve half of the highest value until 52 cycles under the same scanning voltage span. The comparison of the stability of our PeLEDs with those of state-of-the-art is also summarized in Supplementary Table 1, where the much-improved lifetime at such a high electrical excitation represents a notable improvement compared with those of state-of-the-art PeLEDs.

## Recombination kinetics in PeLEDs

To clarify the differences in device performance, particularly at high current densities, we performed a series of optical measurements to investigate the recombination kinetics of perovskites. We conducted transient absorption (TA) and fluence-dependent time-resolved-photoluminescence (TRPL) measurements. The characteristics and relevant 2D pseudo-colour images of TA measurements are shown in Supplementary Fig. 4 and Fig. 2a, b respectively. We observe two primary photobleaching bands situated at 450 nm and 780 nm in both cases, which can be ascribed to the typical sub-band of iodide-perovskites and radiative recombination respectively[38]. Notably, no distinct charge carrier transfers in the different time scales and additional photobleaching features can be found, clearly suggestive of the absence of low-dimensional phases. These findings are in line with the X-ray diffraction (XRD) patterns and grazing-incidence wide-angle X-ray scattering (GIXWAS) measurements, where only the signatures of 3D perovskites were visible (Supplementary Fig. 5).

We then investigated the scaling of the initial PL intensity (at $t = 0$, $I_{PLO}$) with excitation density from TRPL measurements (Supplementary Fig. 6) to uncover the nature of the carrier recombination[24]. As shown in Fig. 2c, the $I_{PLO}$ for both samples exhibit a quadratic dependence on carrier density, suggesting that the photogenerated carriers predominantly undergo bimolecular recombination, typical of free carrier processes. Additionally, we observe that the PL intensity gradually

deviated from the quadratic relationship with the increase of the carrier concentration, indicating a growing prominence of the Auger process[39]. A similar phenomenon is also visible in the evolution of relative PL quantum yields (PLQYs) as a function of carrier density for both cases (Fig. 2d). Notably, the relative PLQYs of FCT-films are much more tolerant to high carrier density compared to F-samples, suggesting a retarded Auger recombination[18,24]. These observations are consistent with the device characteristics, where the peak EQE appeared at high current densities reaching hundreds of mA cm$^{-2}$. In addition, we note that PLQYs of FCT-films are also higher at low carrier densities (<-10$^{15}$ cm$^{-3}$), owing much to a mitigated trap-assisted non-radiative recombination. Such a difference eventually leads to a higher absolute PLQY of 81% for FCT-films compared to 62% of F-films at the best excitation conditions (Supplementary Fig. 7), in line with the higher peak EQE of FCT-devices than F-ones.

Having confirmed that bimolecular recombination is the dominant recombination pathway and the absence of low-dimensional phases, we are able to quantify the recombination kinetics with a combined use of TA and fluence-dependent TRPL by the following equation[40]:

$$\frac{dn}{dt} = -k_1 n - k_2 n^2 - k_3 n^3 \tag{1}$$

where $k_1$ is assigned to the trap-assisted nonradiative recombination constant, $k_2$ is the bimolecular recombination (band-to-band recombination) rate constant, $k_3$ is the trimolecular (Auger) recombination rate constant. Here, $k_1$ was experimentally determined by TRPL spectra at low fluences, while $k_2$ and $k_3$ are deduced from both TRPL and TA tests (Supplementary Figs. 4, 6, 8, and 9). We show the extracted ratios of the various recombination pathways upon carrier densities in Fig. 2e, f and summarized the average recombination constants in Fig. 2g and Supplementary Table 2. The TRPL-derived rates are in good agreement with that extracted from TA measurements. Compared to F-films, we observe largely retarded mono-molecular and Aguer recombination constants but close bimolecular recombination rates in FCT-samples. These contrasts thus lead to a much more dominant role of radiative recombination in FCT-films across a wide range of excitation fluence (Fig. 2e, f). The average $k_3$ of FCT-films extracted by TRPL and TA is $(1.15 \pm 0.64) \times 10^{-28}$ cm$^6$ s$^{-1}$ and $(5.28 \pm 0.43) \times 10^{-29}$ cm$^6$ s$^{-1}$, respectively. Notably, the best sample showed an ultra-slow Auger rate of $1.29 \times 10^{-29}$ cm$^6$ s$^{-1}$ (obtained by TA). These results are among the best so far for lead-halide perovskites and are only one order of magnitude faster than that of III-V emitters (e.g., GaInN, -2 × 10$^{-30}$ cm$^6$ s$^{-1}$)[39,41,42]. The notably suppressed Auger recombination in FCT-films is critical for the mitigated current-efficiency roll-off in the devices. More detailed discussions are provided in Supplementary Note 1.

## Carrier injection kinetics in PeLEDs

Despite the slow-downed Auger recombination in FCT-films with optical excitation, we understand that Augur losses could happen at a low carrier density with electrical excitation, especially in the case of unbalanced charge injections. Given that both devices of interest employed identical charge transport layers, we focused our investigations on the perovskites. Ultraviolet photoelectron spectroscopy (UPS) was used to examine changes in the energy levels of the F- and FCT-films. The results reveal that the introduction of CsTFA causes a shift in the secondary electron cutoff from 17.25 eV to 17.43 eV (Supplementary Fig. 10a). Further calculations indicate that the valence band maximum (VBM) of the perovskite shifts from −5.46 eV to −5.26 eV. The resulting flat-band energy level diagrams are illustrated in Supplementary Fig. 10b, which display that the VBM of the perovskite becomes more closely aligned with the highest occupied molecular orbital (HOMO) level of TFB. Given that the electron mobility of the ZnO electron transport layer is typically much higher

than that of TFB, the improved hole injection facilitates more balanced charge transport.

Another critical observation we noted is the significant difference in morphology between the F- and FCT-films. As illustrated in the SEM and atomic force microscopy (AFM) topographical images (Fig. 3a, b), the dense FCT-films exhibit a small root-mean-square (rms) roughness of 1.67 nm, in stark contrast to the F-films, which display prominent pinholes and an enhanced rms roughness of 5.42 nm. Achieving such a uniform thin film in the FCT-cases is very challenging for 3D perovskite thin-film emitters, considering the ultra-thin-film thickness typically required to enhance charge carrier confinement for efficient recombination.

We realize that the notable changes on film morphology may affect the electrical properties of devices[43]. We thus conducted simulations of the electric potential distribution across simplified device architectures—one featuring densely packed emissive films and the other with discrete perovskite grains (Fig. 3c, d)[44]. The simulations reveal that in both configurations, the electric potential gradient is concentrated near the hole transport layer (TFB)/anode interface, diminishing once moving away from the interfaces (region I and II in Fig. 3c and region III in Fig. 3d). However, it is worthy to mention that excessive holes accumulate at the HTL–ETL interface once increasing the bias, leading to additional charge transport pathways and hence potential electrical shunts (Fig. 3e). Figure 3f displays a further analysis regarding the quantification of charge carrier losses dependent on the surface coverage and applied bias. We find that the electrical shunt can be negligible at low bias even for those with low surface coverage. However, the loss of charge carriers significantly increases once the bias exceeds 4.0 V, suggesting an increasing current-efficiency roll-off over enhancement in bias.

To link the simulation and experimental results, we fabricated devices with a structure of ITO/PEIE-modified ZnO/TFB/MoO$_3$/Au and compared their $J–V$ characteristics with those with perovskite emissive layers (Supplementary Fig. 11). Consistent with our simulation results, we note that the current densities of the devices without perovskites are more than one order of magnitude lower at a small bias (<2.0 V), suggesting that the charge injection barrier between TFB and ZnO is large enough to prevent the shunts at the TFB/ZnO interfaces for PeLED devices. In addition, the current density of the ZnO/TFB-only device quickly rises with increasing the bias and eventually reaches the same order of magnitude as the perovskite device at high voltages. These results align with our simulation results that the electrical shunts can be very severe at the high bias necessary to achieve high radiance. These findings demonstrate that dense film morphology enables minimized detrimental effects such as charge accumulation and electrical shunts, thereby mitigating the issues of current-efficiency roll-off[45,46].

To confirm and generalize our findings, we investigated two additional types of devices with controlled surface coverages of perovskites. One approach was to modify the morphology by varying the quantity of excess FAI into FAPbI$_3$, with the molar ratios of FAI to PbI$_2$ ranging from 2.0:1 to 2.8:1. The other case is to introduce DMF vapour atmosphere during thermal annealing to control the grain growth, using a fixed FAI to PbI$_2$ ratio of 2.0:1[47]. In brief, we find that the films with high surface coverage commonly give rise to enhanced performance at high current densities, thereby supporting above analysis that the dense film morphology is crucial. The corresponding device characteristics and film morphologies are detailed in Supplementary Figs. 12 and 13, and a more thorough description is available in Supplementary Note 2.

## Roles of TFA$^-$ addition

Having concluded that the decent performance of FCT-devices at high current densities mainly stems from the slowed-down Auger recombination rates of perovskites as well as balanced charge injection, the next question that arises is why TFA$^-$ addition can achieve these

 

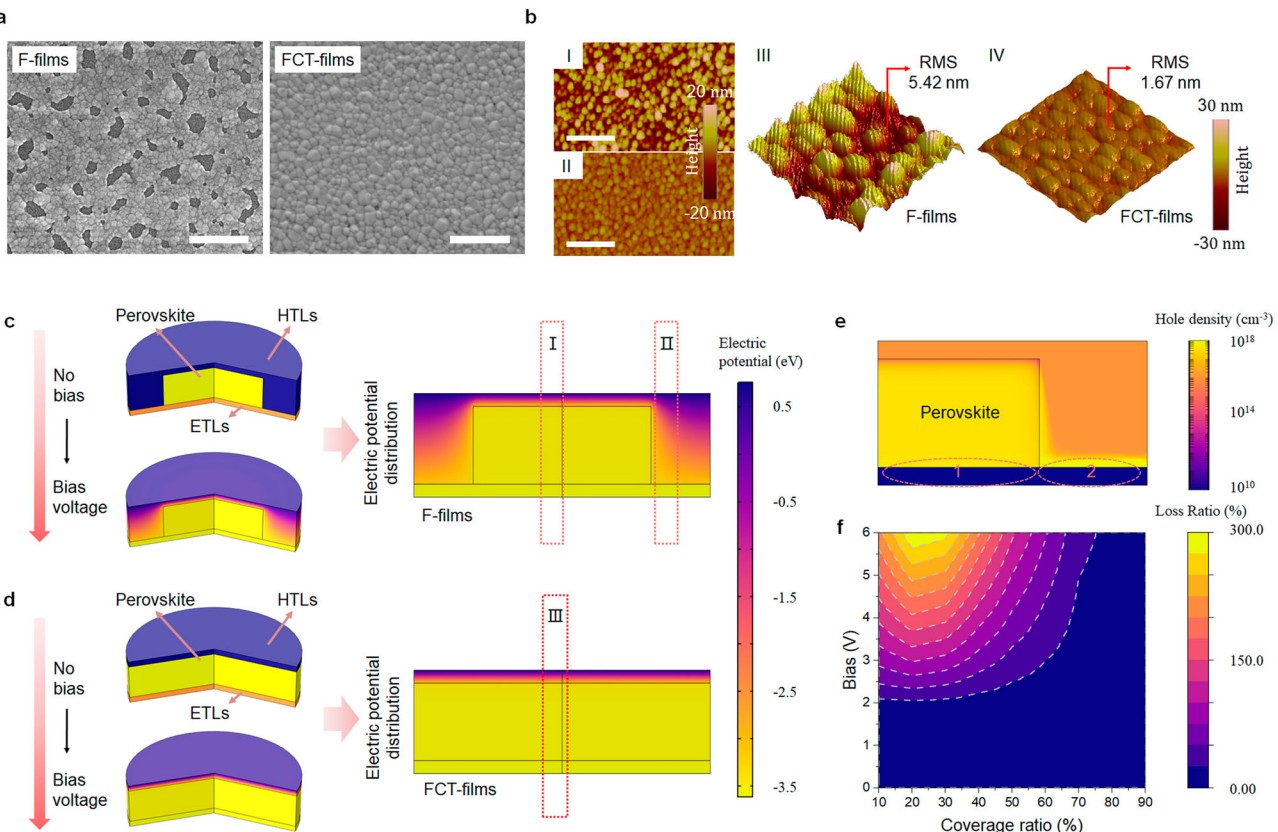

**Fig. 3 | Perovskite film characteristics and device electric simulations. a** SEM images of F- and FCT-films. The scale bar represents 1 μm. **b**, AFM and 3D AFM images of F- and FCT-films. The scale bar represents 1 μm. The electric simulation and simulated electric field profile of devices with low (**c**) and full coverage (**d**) perovskite films (the color represents the electric potential gradient distribution inside of the devices). **e** Simulated holes distribution after carrier injection in PeLEDs at the bias of 6 V. **f** Dependence of loss ratio on bias and surface coverage.

effects. In addition, the observed mitigated trap-assisted recombination caused by TFA⁻ is possible to be correlated with the prolonged operational lifetime. To understand the underlying mechanisms, probing the TFA⁻ interaction with perovskites is crucial. We conducted X-ray photoelectron spectroscopy (XPS) and time-of-flight secondary ion mass spectrometry (ToF-SIMS) measurements, aiming to verify the presence of TFA⁻ within the perovskite films and investigate their distribution perpendicular to the substrate. Different from the F-films, the core-level spectra of C 1s and F 1s, as depicted in XPS images (Fig. 4a), clearly reveal the distinct features corresponding to −O−C=O and −CF₃ functional groups, confirming the presence of TFA⁻ on the surface of the perovskite films. In addition, ToF-SIMS measurements show that the TFA⁻ ions are evenly distributed throughout the perovskite films (Supplementary Fig. 14). These results not only confirm the incorporation of TFA⁻ into the perovskite films but also help to establish the connection to the modified recombination dynamics and film morphology.

Previous reports in II-IV semiconductor nanocrystals suggest that a decouple of wave-function overlap between electrons and holes usually gives rise to a decreased Auger process. Such an effect can be achieved by either constructing a type II core-shell heterojunction or a careful design of organic ligands on the surface[48–50]. Given that TFA⁻ is a strong and well-known electron-withdrawing moiety, we used density functional theory (DFT) calculation to unveil the changes in electron cloud density of crystal surface with and without TFA adsorption. We examined the surface lattice structures of FAPbI₃ and its variant, where one Cs⁺ cation substitutes FA⁺ (denoted as $FA_xCs_{1-x}PbI_3$) for comparison (Fig. 4b). As expected, the strong electron-withdrawing capability of −CF₃ moiety leads to an electron polarization and thus

results in the reduction of electron cloud density in the lattice. This redistribution results in a decrease in wave-function overlap, reducing Coulomb electron-hole interaction and thereby decreasing Auger recombination. Our analysis on temperature-dependent PL enabled us to determine the exciton binding energies (Supplementary Fig. 15), which are 44 meV for F-films and about 28 meV for FCT-films, respectively. The decrease in exciton binding energy often leads to a retarded Auger process due to the changes in Coulomb interactions within the materials. Additionally, the electron localization function (ELF) and the Bader charge analysis (Supplementary Fig. 16) of the samples suggests that the atoms of the FAPbI₃ loss electrons in the presence of TFA. This obviously changed electron distribution of lattices is attributed to large electronegativity of the dangling F atoms in the TFA.

Having revealed the underlying reasons behind the decreased Auger recombination rates, we proceeded to investigate the origins of morphological variation, which is commonly associated with a change of crystallization process. We then performed ¹H nuclear magnetic resonance (¹H NMR) and Fourier-transform infrared spectroscopy (FT-IR) measurements, aiming to gather more information about the chemical interactions between perovskite precursors and TFA⁻ anions, which could be signs of the changes in the formation of intermediates and thus varied crystallization process. In Fig. 4c, we present the changes in the chemical shifts of the active protons in FA⁺ upon the addition of CsTFA (highlighted by the red square). For both cases, the chemical shifts moved toward the low field after TFA⁻ addition in the respective systems (FAI/CsTFA and PbI₂/FAI/CsTFA), confirming the interactions between FA⁺ cations and TFA⁻ anions. This can be assigned to the ionic interactions between positively charged FA⁺ and negatively charged TFA⁻, or additional hydrogen bonding

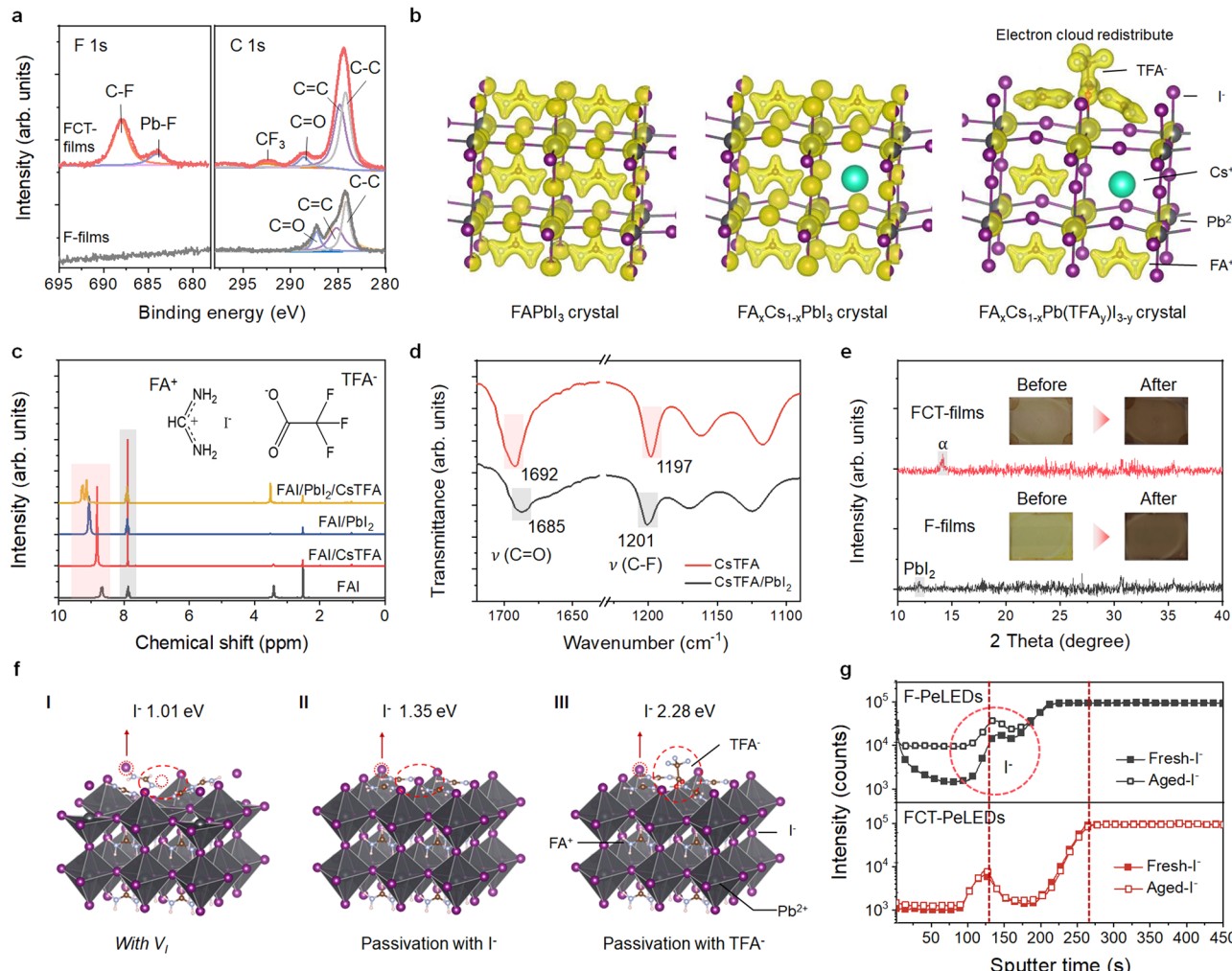

**Fig. 4 | Mechanisms of TFA⁻ anions on improving the PeLED performance.**
**a** Core-level spectra of F 1 s and C 1 s obtained from high-resolution XPS of F- and FCT-films. **b** Lattice structures and corresponding electron cloud density for FAPbI₃, FA₁₋ₓCsₓPbI₃, and FA₁₋ₓCsₓPbI₃ with TFA⁻ adsorption, respectively. The isosurface is 0.1 eV/Å³. **c** ¹H NMR spectra of different materials dissolved in DMSO-d6. The red and grey squares highlight the characteristic peaks of FA⁺. **d** ATR-FTIR spectroscopy data for CsTFA and CsTFA–PbI₂ samples (with molar ratio of 1:1). **e** XRD of the F- and FCT-films before thermal annealing. The insert are the photographs of the perovskite films before and after thermal annealing.
**f** Determination of desorption energy ($E_d$) of the I⁻ on different perovskite surface: With iodide vacancies (I), prefect structures (II), and passivated with TFA⁻ anions.
**g** ToF-SIMS characteristics of F- and FCT-devices before and after electrical aging. Here, the red circle highlights the increased accumulation of I⁻ ions close to the anode for F- devices.

interactions. In addition, the FT-IR results showed that the C=O vibration of TFA⁻ shifts to a lower wavenumber by mixing with PbI₂, accompanied by the shift of the C−F vibration frequency to a higher wavenumber (Fig. 4d). All these observations demonstrated the interactions of TFA⁻ anions with both FA⁺ and Pb²⁺ cations[35].

We found that the additional interactions of TFA⁻ with perovskite precursors not only facilitate the perovskite formation but also homogenize the grain growth. The former is evident from the presence of 3D perovskites within the precursor films (before thermal annealing), as their diffraction (Fig. 4e) and absorption features can be clearly identified by XRD and UV-Vis absorption measurements (Supplementary Fig. 17). In comparison, films without TFA⁻ displayed no prominent perovskite signals, despite the presence of PbI₂ signal. In the insets of Fig. 4e, the digital images visually capture the differences between F- and FCT-precursor films, highlighting the effect of TFA⁻ addition on facilitating perovskite nucleation. We attribute these discrepancies to the reduced formation of lead-Lewis base adducts with TFA⁻ addition, that is, Pb•DMSO in the current case; Pb•DMSO can slow down perovskite crystallization as they require extra energy to release the

coordinating solvents[35]. This speculation can be further evident from FT-IR spectra that the S=O stretching vibration of FAI•PbI₂•DMSO complexes shifts to a higher wavenumber once incorporating TFA⁻ anions, confirming a reduced formation of intermediate FAI•PbI₂•DMSO adducts (Supplementary Fig. 18). Additionally, our dynamic light scattering (DLS) analysis revealed that TFA⁻ addition reduces the size of the halide plumbate colloids, from an average size of 164 nm down to 105 nm (Supplementary Fig. 19) in precursor solutions. This suggests a more uniform nucleation process and a lower total free energy during perovskite phase formation. Taken together, these observations consistently point to a faster and more uniform crystallization process with TFA addition, which is usually beneficial for formation of dense and smooth perovskite films.

Interestingly, although fast crystallization of perovskites commonly leads to high densities of defects[35], we still observed reduced trap-assisted non-radiative recombination rates in FCT-films (Fig. 2g). This contrast suggests an effective capability of TFA⁻ in defect passivation[35,36]. We thus investigated the adsorption geometry of TFA⁻ on vacancy-involved crystal lattice by DFT calculations. Our

calculations indicate that TFA⁻ anions tend to adsorb on the surface of lattice instead of being in the crystals (Supplementary Fig. 20), as a result of their large ionic radius (2.38 Å). Notably, the adsorption energy ($E_{ads}$) of TFA⁻ anion on perovskite crystals (exposed lead cations) is as high as 4.55 eV, much larger than that of the iodide anions (~3.11 eV). This difference indicates a strong tendency of TFA⁻ to work with defect involved perovskite surfaces, and hence the passivation effect can be rationalized.

One consensus is that the reduced formation of vacancy-type defects contributes to the stabilization of ions within perovskite structures. To establish the correlation between ion movement and device stability, we calculated the desorption energy ($E_d$) of surface iodides (I⁻) in different local environment (Fig. 4f). A lower $E_d$ indicates facilitated ion migration and thus accelerated device degradation, and vice versa. For pristine crystals and those with $V_I$, the $E_d$ is 1.01 eV and 1.35 eV, respectively, suggesting that iodide anions adjacent to the defects are more likely to migrate out of the lattice. In contrast, the $E_d$ values significantly increase up to 2.28 eV for the crystal lattice healed by TFA⁻, indicative of a greatly enhanced difficulty for neighboring iodides to become mobile[29].

To experimentally confirm the role of TFA⁻ in suppressing ion migration, we investigated the ion distribution in fresh and aged devices by TOF-SIMS measurements. The aged devices were measured at a fixed current density of 100 mA cm⁻² for 10 h ahead of tests. The comprehensive TOF-SIMS results for the devices of interest are shown in Supplementary Fig. 21. Figure 4g displays the evolution of iodide distribution in F- (upper panel) and FCT- devices (bottom panel). The former exhibits a distinct accumulation of iodides at the interface between HTL and the contacts, whereas the FCT-devices don't show a substantial variation in iodide distribution after aging. These results are consistent with the previous report that halide migration toward the anode is the predominant factor limiting the operational lifetime of PeLEDs[51].

## Discussion

In summary, we achieve highly bright and one of most stable perovskite light-emitting diodes reported to date, overcoming significant limitations of current-efficiency roll-off at high current densities. This breakthrough is achieved by introducing trifluoroacetate anions (TFA⁻) into the three-dimensional structure of perovskites. The most critical effect observed is a substantial reduction in Auger recombination, primarily attributed to decreased electron-hole wave-function overlap. Moreover, we demonstrate that dense and smooth perovskite films are crucial for balancing charge injection at high electrical excitation and thus high brightness. Further studies reveal that the use of TFA⁻ anions in perovskites suppresses halide migration and hence contributes to decent device stability. Our demonstration of high-performance PeLEDs under intense electrical excitation opens possibilities for their use in high-power applications including the development of perovskite laser diodes.

## Methods
### Materials
Formamidinium iodide (FAI) and 5-ammonium valeric acid iodide (5AVAI) were bought from Greatcell solar. Cesium trifluoroacetate (CsTFA) was purchased from Alfa Aesar. Other chemicals were obtained from Sigma−Aldrich. All materials were used without extra treatments. Zinc oxide (ZnO) was synthesized based on previous reports[2,7–9]. 2.8 mmol tetramethylammonium hydroxide pentahydrate (TMAH·5H₂O) in 5 mL ethanol was added into 1.5 mmol zinc acetate hydrate (Zn(Ac)₂·2H₂O) in 15 mL dimethyl sulfoxide (DMSO). Then, the solution was stirred at 40 °C for 3 h to form ZnO nanoparticles. The obtained ZnO was washed with ethanol and ethyl acetate for 2 times, and finally dispersed in ethanol (8 mL).

### Perovskite precursors
0.15 mmol CsTFA, FAI, lead iodide (PbI₂), and 5AVAI ($n$(CsTFA):$n$(FAI):$n$(PbI₂):$n$(5AVAI) = 0.12:1.28:1:0.1) were dissolved in N, N-dimethylformamide:dimethyl sulfoxide (DMF:DMSO = 9:1) mixture to form FCT-precursors. 0.15 mmol FAI, PbI₂, and 5AVAI were dissolved in DMF:DMSO (9:1) mixture at a molar ratio of 1.4:1:0.3 to form F-precursors. 0.13 mmol FAI, PbI₂, and 5AVAI are dissolved in DMF at a molar ratio of (2.8, 2.4, or 2):1:0.15 to form FAI-2.8, FAI-2.4, and FAI-2.0 based precursors. The precursor solution was stirred overnight and filtered with polytetrafluoroethylene filters (0.22 μm) before use.

### Device fabrication
ZnO was spin-coated onto the precleaned indium tin oxide (ITO) substrates at 5000 r.p.m for 30 s, and annealed at 120 °C for 10 min in air. Polyethylenimine ethoxylated (PEIE) in isopropanol (1.5 mg mL⁻¹) was spun on ZnO at 5000 r.p.m. for 30 s, and annealed at 100 °C for 10 min. Then, the perovskite precursor was deposited onto the ZnO/PEIE layers at 500 r.p.m. for 3 s and 6000 r.p.m. for 30 s, respectively. The FCT- and F-precursor films were annealed for 15 min at 90 °C and 100 °C, respectively. After cooling, phenethylammonium iodide (PEAI, 0.25 mg mL⁻¹ in isopropanol) was dynamically spun on the perovskite layer at 4000 r.p.m for 30 s. Next, poly(9,9-dioctyl-fluorene-co-N-(4-butylphenyl)diphenylamine) (TFB, 12 mg mL⁻¹ in chlorobenzene) was spun at 3000 r.p.m. for 45 s. Finally, 6 nm of molybdenum oxide (MoO₃) and 100 nm of gold (Au) were sequentially deposited by thermal evaporation. For optimized FAI-2.8, FAI-2.4, and FAI-2.0 based devices: ZnO was spin-coated onto ITO at 4000 r.p.m for 30 s without annealing. Then, PEIE (1.2 mg mL⁻¹ in IPA) was deposited on ZnO at 5000 r.p.m. for 30 s, and annealed at 100 °C for 10 min. The perovskite precursor was spun onto ZnO/PEIE at 500 r.p.m. for 3 s and 4000 r.p.m. for 30 s, respectively. The precursor film was annealed at 100 °C for 10 min.

### Characterizations
PeLEDs measurements were conducted using a Keithley 2400 source meter and an integration sphere equipped with a QE Pro spectrometer. The stability test was taken on the same test platform in a nitrogen-filled glovebox with a constant room temperature of 20 °C. SEM was measured using a JEOL JSM-7500F microscope. AFM was performed using a Bruker Dimension Icon microscope. FTIR was tested using a commercial FTIR testing equipment. XRD was tested using a X-ray diffractometer (Panalytical X'Pert Pro) with an X-ray tube (Cu Kα, λ = 1.5406 Å). XPS was measured using a Scienta ESCA 200 spectrometer with a monochromatic Al (Kα) X-ray source. ToF-SIMS tests were conducted using a ToF-SIMS.5 instrument from IONTOF, operated in the spectral mode using a 25 keV Bi³⁺ primary ion beam with an ion current of 0.78 pA. ¹H NMR was collected using a Bruker Ultra Shield Plus 400 MHz NMR system. PLQYs was recorded using a Quanta-Phi integrating sphere with a Fluorolog system under the excitation wavelength of 400 nm. TA measurements were performed using a Helios setup. The transient dynamics in fs-ns time region (50 fs - 7 ns) was acquired by Helios that works in a nondegenerate pump−probe configuration. The pump pulses were generated from an optical parametric amplifier (OPerA Solo) that was pumped by a 1-kHz regenerative amplifier (Coherent Libra, 800 nm, 50 fs, 4 mJ). A mode lock Ti-sapphire oscillator (Coherent Vitesse, 100 fs, 80 MHz) was used to seed the amplifier. The probe pulse was a white light continuum generated by passing the 800 nm fs pulses through a 2 mm sapphire plate for visible part (420−780 nm).

### Device simulation
Perovskite devices were modelled with the software of COMSOL Multiphysics 6.0 and the corresponding Semiconductor Module, the main method of which is based on solving the electrostatic equation and the drift−diffusion equations for electrons and holes. 2D

axisymmetric model was used to evaluate the influence of surface coverage in 3D situation by reducing the calculation cost in the 3D model. Electrical potentials were calculated through Poisson equation by taking both carrier dynamics (recombination and transport) and electrostatics into consideration. The detailed model is based on the as-mentioned device consisting of ITO/ZnO/PEIE/perovskite/TFB/MoO$_3$/Au, where TFB and PEIE-modified ZnO are defined as HTLs and ETLs, respectively. The coverage ratio is defined as the area ratio of Region I (perovskite area) and Region I + Region II (perovskite area and transport contact) while the loss ratio is taken as the area-averaged product ratio of electrons and holes in the area without perovskite (Region II in Fig. 3c) to that in the region with perovskite coverage (Region I in Fig. 3c) to estimate the possible loss at the interface through the charge accumulation. The loss ratio can also be affected by the specific interfacial recombination coefficients, and it is not included in the discussion because of our focus on the impact of perovskite coverage on the device performance.

$$\text{Loss Ratio} = \frac{(\int_{\text{region II}} n \times p)/\text{area}_{\text{region II}}}{(\int_{\text{region I}} n \times p)/\text{area}_{\text{region I}}} \times 100\% \qquad (2)$$

### First-principles calculation

DFT calculations were conducted in the framework of the density functional theory using CP2K package (version 7.1) with a plane-wave density cutoff of 500 Ry. The DFT-D3 approach was employed to account for the electronic exchange-correlation interaction in the presence of the organic molecules. CP2K with BFGS scheme was employed to fully relax the crystals structures, and the force convergence criterion was settled to $4.5 \times 10^{-4}$ hartree/bhor. Goedecker-Teter-Hutter (GTH) pseudopotentials were used to describe atomic core electrons, and the valence electron orbitals were expanded into DZVP-MOLOPT-SR-GTH basis sets. For the geometry optimization, the system multiplicity was relaxed until converged.

## Data availability

The published article includes all data analyzed and necessary to draw the conclusions of this study in the figures and tables of the main text and Supplemental Information. Further information and requests should be directed to the corresponding authors.

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

## Acknowledgements

This work was supported by the National Natural Science Foundation of China (Grants 52250060, 62274135, 62288102, 52302167, 62175048), Key project of Ningbo Natural Science Foundation (Grants 20221JCGY01049). F.G. is a Wallenberg Scholar and acknowledges the financial support from the Swedish Strategic Research Foundation (SIP21-0151), the European Research Council Consolidator Grant (LEAP, 101045098), the Olle Engkvists Stiftelse, and the Swedish Government Strategic Research Area in Materials Science on Functional Materials at Linköping University (faculty grant SFO-Mat-LiU no. 2009-00971). G. Li acknowledges the financial support from the Research Grants Council of Hong Kong (GRF Grant Nos. 15221320, CRF C7018-20G, C4005-22Y), the Shenzhen Science and Technology Innovation Commission (Project No. JCYJ 20200109105003940), the Hong Kong Innovation and Technology Commission (GHP/205/20SZ), the Hong Kong Polytechnic University (the Sir Sze-yuen Chung Endowed Professorship Fund (8-8480), PRI strategic Grant (1-CD7X), RISE Strategic Grant (Q-CDBK)). M. Li acknowledges the financial support from the Research Grants Council of Hong Kong (Project No. 25301522, 15301323, 15300824, C5003-24E), National Natural Science Foundation of China (22373081).

## Author contributions

F. G. and W. X. conceived the idea and supervised the project; Z. L., Q. W., and W. G. performed the experiments and analyzed the data; Y. W. performed device simulation and analyzed the data; Q. W. performed transient absorption under the supervision of M. L.; C. T., Y.Z. and Z. W. L. contributed to the data analysis. G. L. supervised Z. L. and contributed to the interpretation of results; W. X., F. G., and Z. L. wrote the manuscript; X. L. and Z. W. provided revisions to the manuscript. All authors discussed the results and commented on the manuscript.

## Funding

## Competing interests

The authors declare no competing interests.
