## [Transparent Peer Review file · Nature Communications]

Highly bright perovskite light-emitting diodes enabled by retarded Auger recombination

Corresponding Author: Professor Feng Gao

Version 0:

Reviewer comments:

Reviewer #1

(Remarks to the Author)

In this work, the authors utilized CsTFA as an additive to suppress Auger recombination in 3D perovskite emitters, realizing perovskite LED with a peak EQE of 21.4% at 783 nm and a maximum radiance of $2409 \text{ W Sr}^{-1} \text{ m}^{-2}$. Although the radiance of devices is high, the paper is poorly written with a numerous of mistakes and the characterizations are incomplete or incorrect. More importantly, the mechanism of CsTFA suppressing Auger recombination has not been thoroughly discussed, and CsTFA has been widely used in perovskite preparation process for enhancing perovskite film' quality. Therefore, I do not recommend the publication of this manuscript on Nature Communications. Below are some additional and more specific comments for the authors.

1. The mechanism of CsTFA suppressing Auger recombination should be thoroughly clarified. The reviewer suggest using cesium acetate as a control both experimentally and theoretically to verify the effect of electron-withdrawing moiety on suppressing Auger recombination.
2. In Fig. 2, the transient absorption spectroscopy (TA) was not fully tested. The reviewer are concerned about the accuracy of K3 value extracted from such incomplete TA spectra. Please provide or retest the complete TA spectra.
3. Although Fig. 3a shows a reduction in the porosity of the perovskite films, there was no significant change observed in the grain size or uniformity between FCT-films and F-films. Please provide evidence of enhanced crystallinity and statistical analysis of grain size distribution.
4. In Fig. 4b, it is difficult to observe the changes in surface charge density of the crystal surface. The reviewer think it is necessary to conduct a more in-depth investigation into the interaction between TFA and perovskite, and provide more direct evidence to demonstrate the changes in surface charge density.
5. The absorption spectra in Fig. S13 look quite abnormal. Please also retest the absorption measurements.
6. In Fig. 1a, please provide a high-resolution cross-sectional SEM image. Does the thickness of perovskite film change after the addition of CsTFA?
7. The reviewer think it is not strict to simply attribute the reason for balanced charge injection in CsTFA-LEDs to the dense film morphology considering that the energy level of perovskite film may be changed by adding CsTFA. Please provide comprehensive characterization and analysis.
8. It is assumed, as the author claimed, that for LEDs with poor coverage of perovskite film excessive holes accumulated at the perovskites edges and at the HTL-ETL interface under increased bias, then why does hole injection become more difficult in J-V characteristic?
9. In Fig. 4g, Why does the best FCT-PeLEDs show a distinct accumulation of I at the interface in ToF-SIMS measurement?
10. There are a numerous of errors in this manuscript, to name but a few:
 - 1) Line 83, the sample name is incorrectly labeled.
 - 2) Lines 170-175, the description is inconsistent with the labelled in Fig. 3b.
 - 3) There is an issue with the image in Fig. S1.

Reviewer #2

(Remarks to the Author)

This manuscript reports high efficiency NIR PeLEDs at high brightness. By incorporating CsTFA into the perovskite, the

Auger recombination rate is reduced, thus the efficiency roll-off of PeLED under high current density is reduced. The reduction of Auger recombination was proved by TA, TRPL and other tests, and the explanation of the mechanism is fairly clear. The causes and effects of TFA on the morphology of perovskite films were also revealed. The efficiency of this PeLED under high current is indeed remarkable. But we still found some problems in this work, and we thought that this work could only be published if the author solved the problems below.

1 , According to the theory put forward by the author, "In the configurations with poor coverage, additional pathways for charge transport are evident between ETLs and HTLs, suggesting potential electrical shunts", at low current density, the phenomenon of electrical shunts also exists and will reduce the EQE. But why in the comparison experiments (Supplementary Fig. 10 and 11), under low current density and low voltage, the EQE of devices with low surface coverage is higher and the opening voltage is lower?

2 , This article mainly explains the role of TFA- anions, but does not mention the role of Cs+ cations. If Cs+ ion does not affect the device performance, why not use FATFA, but CsTFA, which is different from the main cation of perovskite?

3 , Is the author's interpretation of Supplementary Fig. 12 and Figure 4a inverted? XPS is a surface test that cannot indicate the presence of TFA- inside the perovskite.

4 , In the XPS test, the author proves the existence of TFA- by the appearance of $-O-C=O$ characteristic peak, but in the components of perovskite, 5AVAI also contains $-O-C=O$, thus the author needs to avoid the influence of 5AVAI to show that this characteristic peak comes from TFA-.

Reviewer #3

(Remarks to the Author)

Li et al. have demonstrated perovskite LEDs that maintain a high external quantum efficiency (EQE) of over 20% at current densities exceeding 2 A/cm^2 . This work represents a significant milestone in the perovskite LED field and highlights the potential of perovskite LEDs for practical use. The reviewer strongly recommends the publication of this work without any reservations.

Version 1:

Reviewer comments:

Reviewer #1

(Remarks to the Author)

In the revised manuscript, the authors have supplemented additional experiments and discussion to improve the manuscript. They addressed most questions raised in the first round of revision, and detailed explanations were included in the response letter. In particular, the response and explanation to the mechanism of balanced charge injection were thoroughly discussed. However, the mechanism on retarded Auger recombination should be further strengthened. Therefore, the authors still need to address the following comments.

1. Although the changes on electron density are highlighted in Fig. 4b, the analysis of charge distribution is still qualitative and unclear. It is also confusing that all charge around I ions are disappeared after TFA anions adsorbed on the perovskite surface. From the results shown in Fig. 4b, TFA can exhibit a large interaction range over 10 \AA , is this reasonable? Please provide more convincing evidence and a more detailed analysis of the effective interaction range of TFA.

2. Also in Fig. 4b, it is recommended to provide direct evidences, such as on Bader charge analysis or electron localization function(ELF) to show the quantitative changes of charge quantity. Besides, the legend is missing.

3. The author should provide the specific values of the exciton binding energies.

Reviewer #2

(Remarks to the Author)

The authors have made detailed response to the reviewers' comments. The revision is greatly improved in the current form. So I think this work is suitable for publication in Nature Commu now.

Version 2:

Reviewer comments:

Reviewer #1

(Remarks to the Author)

The authors have well addressed the reviewers' comments, and thus I recommend its publication on Nature Communications as is.

We sincerely acknowledge the editor and all reviewers for their valuable feedback, which has provided us the opportunity to improve the quality of our manuscript. We are grateful to reviewers 2 and 3 for recognizing the significance of our results and considering them a milestone in the development of perovskite LEDs. We also thank reviewer 1 for the constructive comments. Below, we have provided a point-by-point response and have made the corresponding revisions in the manuscript.

REVIEWER COMMENTS

Reviewer #1 (Remarks to the Author):

In this work, the authors utilized CsTFA as an additive to suppress Auger recombination in 3D perovskite emitters, realizing perovskite LED with a peak EQE of 21.4% at 783 nm and a maximum radiance of 2409 W Sr⁻¹ m⁻². Although the radiance of devices is high, the paper is poorly written with a numerous of mistakes and the characterizations are incomplete or incorrect. More importantly, the mechanism of CsTFA suppressing Auger recombination has not been thoroughly discussed, and CsTFA has been widely used in perovskite preparation process for enhancing perovskite film' quality. Therefore, I do not recommend the publication of this manuscript on Nature Communications. Below are some additional and more specific comments for the authors.

Reply: We appreciate the referee's attention to detail in pointing out some typographical errors in our manuscript. In order to address the concerns from the Referee, we have now included additional experimental results that further support our findings and reinforce the validity of our conclusions.

While it is true that CsTFA has been previously studied as an additive for enhancing the quality of perovskites, prior research has primarily focused on the role of TFA anions in reducing grain size and as ligands for passivating perovskite quantum dots (e.g., Yang et al., Nat. Commun., 10, 665 (2019)). To our knowledge, no previous studies have explored their potential in mitigating Auger recombination in perovskites. In our work, we introduced TFA anions into well-passivated 3D perovskites, which led to significantly enhanced performance

at high current densities. Our results highlight the unique advantages of perovskite LEDs over other solution-processed LEDs and pave the way for future investigations into their potential as laser diodes. As such, we believe our results are of critical importance for the perovskite and LED communities.

1. The mechanism of CsTFA suppressing Auger recombination should be thoroughly clarified. The reviewer suggests using cesium acetate as a control both experimentally and theoretically to verify the effect of electron-withdrawing moiety on suppressing Auger recombination.

Reply: We appreciate the referee's valuable suggestion. In response, we fabricated devices with cesium acetate (CsAc). The J-EQE and J-V-R characteristics for representative devices are presented in the following figures. However, we observed a significant decline in LED performance compared to even the control devices without any additives, with a peak EQE of 13.2% and a moderate radiance of $\sim 700 \text{ W sr}^{-1} \text{ m}^{-2}$. The poor performance discourages further detailed investigations, especially considering that carboxylic acid is known to have only weak electron-withdrawing properties.

Fig. R1. Performance of PeLEDs. a, EQE plotted against current density curves. **b,** Dependence of current density and radiance on voltage.

2. In Fig. 2, the transient absorption spectroscopy (TA) was not fully tested. The reviewers are concerned about the accuracy of K_3 value extracted from such incomplete TA spectra. Please provide or retest the complete TA spectra.

Reply: We thank the reviewer for this constructive comment. We provide the full transient absorption (TA) spectra of the samples in **Fig. R2** and **R3**. The TA plots for each perovskite

were measured from two different test windows, and the breakpoints in the plots are a consequence of the limitations of the instrument.

To accurately compare the Auger recombination rates between the control and target samples, the k_3 values in this work were extracted from the kinetics probed at the photoinduced bleach (PB) band. Such measured decay kinetics is a combination of non-radiative and radiative recombination processes. In general, there is no change in the decay kinetic curve (in the PB band region) when the detection wavelength is altered because all the curves originate from one same PB band. The main reasons for the kinetics changing with wavelength include carrier thermalization and hot carrier cooling, but the timescales for these processes are typically only a few picoseconds.

In order to address the concerns raised by the reviewer, we conducted a comparative analysis of the PB band kinetic curves of the control and target perovskites, as illustrated in **Fig. R4**. Our findings indicate that there is no discernible difference in the kinetic curves in the vicinity of the PB band, thereby substantiating the accuracy of our methodology for extracting the Auger recombination rate.

Fig. R2. TA spectra for (a) F- and (b) FCT-films, respectively.

Fig. R3. TA spectra of F- (a) and FCT-(b) perovskite films with 3.10 eV photons excitation.

Fig. R4. The kinetics extracted from F- (a) and FCT- (b) perovskites, respectively.

3. Although Fig. 3a shows a reduction in the porosity of the perovskite films, there was no significant change observed in the grain size or uniformity between FCT-films and F-films. Please provide evidence of enhanced crystallinity and statistical analysis of grain size distribution.

Reply: Following the suggestion from the Referee, we presented the statistical analysis of grain size distribution by further analyzing the SEM images: the FCT-perovskites exhibit an average grain size of approximately 150 nm, larger than the ~120 nm observed in the F-films (Fig. R5). While the grain size is increased, we find that the crystallinity is negligibly changed. We conducted grazing-incidence wide-angle X-ray scattering (GIWAXS) measurements (Supplementary Fig. 5b) on the F- and FCT-films. Both XRD (Supplementary Fig. 5a) and GIWAXS results show negligible changes in crystallinity for the FCT-films.

Supplementary Figure 5. X-ray diffraction (XRD) patterns and grazing incidence wide angle X-ray scattering (GIWAXS) measurements for the perovskite films. a, XRD images of F- and FCT-films. α and * denote the identified diffraction peaks corresponding to the α -FAPbI₃ and ITO. b-c GIWAXS images of F- and FCT-films.

Fig. R5. The grain size distributions of the F- and FCT- perovskite crystals.

4. In Fig. 4b, it is difficult to observe the changes in surface charge density of the crystal surface. The reviewer think it is necessary to conduct a more in-depth investigation into the interaction between TFA and perovskite, and provide more direct evidence to demonstrate the changes in surface charge density.

Reply: We thank the referee for raising this question which makes us realize that our data could be better presented. We agree that from the first glance it is not straightforward to distinguish the differences on electron density in Fig. 4b. We now have reorganized the images and highlighted the critical changes on electron density with TFA anions adsorbed on the surfaces of perovskites. Compared to those with clean surfaces, the electron cloud density of targeted cases redistributed and somehow polarized toward TFA due to its strong electron-withdrawing ability. It hence leads to a reduction of electron cloud density on perovskites. Such a polarization is correlated with a reduced wave-function overlap, and thus retarding the Auger recombination

rates. Similar reports can be found elsewhere (*J. Phys. Chem. Lett.* **9**, 2098-2104 (2018). *Chem. Mater.* **25**,1318-1331 (2013). *Annu. Rev. Phys. Chem.* **65**, 317-339 (2014)). As such, our calculation results agree well with previous understanding in cadmium-based semiconductors and our experimental results regarding recombination dynamics, although we are not aware of any techniques that can provide direct evidence for the changes on the surface charge density in this case.

Fig. 4b. Lattice structures and corresponding calculated electron cloud density for FAPbI₃, FA_{1-x}Cs_xPbI₃, and FA_{1-x}Cs_xPbI₃ with TFA⁻ adsorption, respectively.

5. The absorption spectra in Fig. S16 look quite abnormal. Please also retest the absorption measurements.

Reply: We thank the referee for this valuable comment. In Supplementary Fig. 15a, we presented the absorption spectra of the F- and FCT-perovskite precursor films before thermal annealing. Due to the presence of solvent residues and the ongoing drying process during the measurements, the absorption spectra show slightly enhanced background signals, likely caused by changes in optical scattering over time. Such an issue is widely visible in measuring precursor films with solvent residues (e.g. *Nat. Commun.* 2019, 10, 2818.). However, we still believe that this data is reasonable, and our conclusion that perovskite crystals had already formed in the FCT precursor films is convincing. This is further supported by the XRD results (Fig. 4e).

To prevent any potential misunderstandings for readers, we have also provided the absorption spectra of the F- and FCT-perovskite films in Supplementary Fig. 15b.

Supplementary Figure 15. Absorption measurements of perovskite films before annealing.

UV-Vis absorption of the F- and FCT-perovskite precursor films before (a) and after (b) annealing. The absorption spectra show the incorporation of TFA⁻ into perovskite precursors facilitated the perovskite formation before annealing.

6. In Fig. 1a, please provide a high-resolution cross-sectional SEM image. Does the thickness of perovskite film change after the addition of CsTFA?

Reply: We have added the high-resolution cross-sectional SEM image for the FCT-perovskite film in Fig. 1a. From the cross-sectional SEM images, we can find that the thickness of FCT-perovskite films is about ~80 nm. Fig. R6 exhibits the cross-sectional SEM images of F-PeLEDs. The thickness of the F-perovskite films is about 60~80 nm, which is similar to the FCT- perovskite films.

Fig. 1a. Configuration and the high-resolution cross-sectional SEM image of FCT-PeLEDs.

The scale bar represents 100 nm.

Fig. R6. The high-resolution cross-sectional SEM image of F-PeLEDs. The scale bar represents 100 nm.

7. The reviewer think it is not strict to simply attribute the reason for balanced charge injection in CsTFA-LEDs to the dense film morphology considering that the energy level of perovskite film may be changed by adding CsTFA. Please provide comprehensive characterization and analysis.

Reply: We are grateful to the referee for raising this constructive comment which significantly helped us to improve the quality of the manuscript. We evaluate the possible shifts of energy levels of the perovskites by performing ultraviolet photoelectron spectroscopy (UPS) measurements, which are displayed in Supplementary Fig. 10a in the revised Supplementary Information. We noted an obvious shift in secondary electron cut-off toward higher binding energy in FCT-films compared to that of F-samples. The valance bands of perovskites are thus determined to be -5.28 eV and -5.46 eV for FCT- and F- films respectively ($\text{He-I}\alpha = 21.22$ eV). Considering that the bandgap of FCT- and F-films are calculated to be ~ 1.49 eV and ~ 1.51 eV according to the UV-Vis absorption, the relevant energy levels of conduction bands are -3.77 and 3.95 eV, respectively.

We show the schematics of the flat-band energy level diagram of the device components in Supplementary Fig. 10b. It clearly indicates that the energy barriers between perovskites and hole transport interlayer (TFB) is slightly reduced in FCT-films, leading to a perfect energy level alignment. As there is no energy barrier of electron injection for both F- and FCT- devices, we believe the facilitated hole injection may contribute to the balanced charge injection and

hence less distinct current-efficiency roll-off.

We have added more descriptions about the effect of energy levels in this manuscript.

Supplementary Figure 10. Determining the evolution of energy levels of perovskites. a, UPS spectra of the F- and FCT- perovskite films (showing the secondary electron cut-off ($E_{\text{cut-off}}$) and the ionization edge. The binding energy is referenced with respect to the Fermi level of the system. $\text{He-I}\alpha = 21.22$ eV). **b,** The flat-band energy level diagram of the F- (left) and FCT- (right) device components. The energy levels of TFB and ZnO are from literatures. The energy levels of conduction bands of perovskites were determined by UPS and optical bandgap from UV-Vis absorption spectra in Supplementary Figure 15b.

8. It is assumed, as the author claimed, that for LEDs with poor coverage of perovskite film excessive holes accumulated at the perovskite edges and at the HTL-ETL interface under increased bias, then why does hole injection become more difficult in J-V characteristic?

Reply: The low-coverage case shows additional charge transport paths (TFB/ZnO and TFB/Perovskite/ZnO, top panel in Fig. 3c and d) between ETLs (PEIE-modified ZnO as ETLs) and HTL(TFB). In this scenario, holes increasingly pile up on the side of the perovskites as well as at the interface between HTLs and ETLs over enhancing the bias (Supplementary Fig. 11). Given that the large electron injection barrier between ETLs and HTLs is possible to inhibit current shunts to some extent, the leakage current of TFB/ZnO usually is small, which make little contribution to current density.

To further clarify the carrier loss of device with low surface coverage, we calculated the relationship between loss ratio and surface coverage, which is depicted in Fig. 3e in the revised manuscript. From the loss ratio spectra, we find that the loss ratio is small for the devices with

low coverage under low bias. However, the loss ratio of low-coverage devices obviously increases once the bias exceeds 4.0 V.

It is known that electron injection from the PEIE-modified ZnO is more efficient than hole injection from TFB (Nat. Photon. 14, 215–218 (2020), Adv. Funct. Mater. 30, 1910464 (2020), ACS Appl. Mater. Interfaces 13, 28546–28554 (2021)). Therefore, hole injection in the active contact (section 1) become more difficult and current density is limited by hole injection from TFB to perovskite.

Fig. 3 c, d, The electric simulation and simulated electric field profile of devices with low (c) and full coverage (d) perovskite films (The color represents the electric potential gradient distribution inside of the devices). **e**, Simulated holes distribution after carrier injection in PeLEDs. **f**, Loss ratio of carrier of devices with bias and coverage.

Supplementary Figure 11. The evidence for the electrical shunts under high excitation. Current density vs voltage curves of the FA-2.4 device and ITO/ZnO/PEIE/TFB/MoO₃/Au device.

9. In Fig. 4g, why does the best FCT-PeLEDs show a distinct accumulation of I at the interface in ToF-SIMS measurement?

Reply: We are grateful to the Referee for this valuable comment. Although we do not have a solid conclusion, we speculate that during thermal evaporation, some of iodides may diffuse toward the electrodes upon heating. Considering the discrepancies of ion diffusion in different mediums, it usually leads to a mild elemental accumulation at the interfaces. However, it should be noted that the halide content at HTM/Au interface is very little compared to that at perovskite/HTM interfaces. Similar phenomenon is visible elsewhere, e.g. *Matter* 4, 3710-3724 (2021), *Small Struct.* 3, 2200063 (2022), *Adv. Funct. Mater.* 33, 2305423 (2023), *Adv. Mater.* 2313981 (2024)). Here, we show one example (**Fig. R7**) reported in *Matter* 4, 3710-3724 (2021) as follows.

[Figure redacted]

Fig. R7 ToF-SIMS depth profiling conducted on NMAI devices (fresh, half-degraded, recovered, and completely degraded) from previous paper (*Matter* 4, 3710-3724 (2021)).

10. There are a numerous of errors in this manuscript, to name but a few:

- 1) Line 84, the sample name is incorrectly labeled.
- 2) Lines 181-187, the description is inconsistent with the labelled in Fig. 3b.
- 3) There is an issue with the image in Fig. S1.

Reply: Many thanks for your advice. Accordingly, we have checked the paper and corrected them in the manuscript. Our changes to the manuscript are given in **blue** text. For example:

- 1) Line 84: Unless otherwise stated, we refer to the samples without and with CsTFA as F- and FCT-films/devices, respectively.

2) Lines 180-186: Another critical observation we noted is the significant difference in morphology between the F- and FCT-films. As illustrated in the field-emission scanning electron microscopy (FE-SEM) and atomic force microscopy (AFM) topographical images (Fig. 3a and 3b), the dense FCT-films exhibit a small root-mean-square (rms) roughness of 1.67 nm, in stark contrast to the F-films, which display prominent pinholes and an enhanced rms roughness of 5.42 nm. Achieving such a uniform thin film in the FCT-cases is notably challenging for 3D perovskite thin-film emitters, considering the ultra-thin film thickness typically required to enhance charge carrier confinement for efficient recombination.

3) We have corrected the Fig. S1 (Fig. S2 in the manuscript).

Reviewer #2 (Remarks to the Author):

This manuscript reports high efficiency NIR PeLEDs at high brightness. By incorporating CsTFA into the perovskite, the Auger recombination rate is reduced, thus the efficiency roll-off of PeLED under high current density is reduced. The reduction of Auger recombination was proved by TA, TRPL and other tests, and the explanation of the mechanism is fairly clear. The causes and effects of TFA on the morphology of perovskite films were also revealed. The efficiency of this PeLED under high current is indeed remarkable. But we still found some problems in this work, and we thought that this work could only be published if the author solved the problems below.

1, According to the theory put forward by the author, "In the configurations with poor coverage, additional pathways for charge transport are evident between ETLs and HTLs, suggesting potential electrical shunts", at low current density, the phenomenon of electrical shunts also exists and will reduce the EQE. But why in the comparison experiments (Supplementary Fig. 11 and 12), under low current density and low voltage, the EQE of devices with low surface coverage is higher and the opening voltage is lower?

Reply: We appreciate the referee for these valuable comments. We agree that the devices with

poorer coverage of perovskites show higher EQEs at low current density regions in both Supplementary Fig. 12 and 13. However, we believe these effects are more reasonable to be attributed to the reduced trap-assisted non-radiative recombination instead of electrical shunts.

For the experiments described in Supplementary Fig. 12, we modified the feed ratio of FAI (FAI: $\text{PbI}_2 = 2.8:1, 2.4:1$ and $2.0:1$) to modify the grain size and film morphology. However, it inevitably brings about differences in perovskite quality, that is, the one with a higher content of FAI leads to fewer defects due to the passivation effects of excess FA^+ and I^- (e.g., *Adv. Funct. Mater.* 30, 1906875(2020)). Similarly, in Supplementary Fig. 13, the solvent annealing treatment leads to larger grain sizes and hence less grain boundary and point defects (*Adv. Mater.* 2014, 26, 201401685. *Nano Energy.* 2016, 28, 417-425).

Supplementary Figure 12. Correlation between surface coverage and device performance.

a, SEM images of FAI-2.8, FAI-2.4, and FAI-2.0 perovskite films. The scale bar represents 1 μm . b, The schematic diagram of device architecture. c-e, Characteristics of PeLEDs with different FAI contents: EQE vs voltage curves (c), EQE vs current density curves (d), and current density/radiance vs voltage curves (e).

Supplementary Figure 13. Understanding the role of perovskite film morphology on device performance. SEM images of FAI-2.0 perovskite films without (a) and with (b) solvent annealing. The scale bar represents 500 nm. c, The schematic diagram of device coverage. d, EQE vs current density curves of FAI-2.0 perovskite films without and with solvent annealing. d, Current density/radiance vs voltage curves of FAI-2.0 perovskite films without and with solvent annealing.

Fig. 3f displays a further analysis regarding the quantification of charge carrier losses dependent on the surface coverage and applied bias. It is reasonable to find that the electrical shunt can be negligible at low bias even for those with low surface coverage, given that most of efficient PeLEDs are all based on nano-island structures. However, the loss of charge carriers significantly increases once the bias exceeds 4.0 V, suggesting an increasing current-efficiency roll-off over enhancement in bias.

Fig. 3f, Dependence of carrier loss ratio on bias and surface coverage. The loss ratio is calculated as the ratio of the electron-hole product in the perovskite-covered region (e.g., Region I in Fig. c) to that in the uncovered region (e.g., Region II in Fig. c).

To experimentally clarify the difference of the possible electrical shunts at the TFB/ZnO:PEIE interface under high or low voltage excitation, we fabricated ZnO/TFB only devices with the structure of ITO/PEIE-modified ZnO/TFB/MoO₃/Au. As shown in Supplementary Fig. 11, A big current difference is observed under the low operating voltage (< 3V). Meanwhile, only a negligible current density is injected into the TFB device under the general operating condition of FAPbI₃ devices (around 2V), indicating efficient blocking of leakage current at the TFB/ZnO:PEIE interface. Therefore, the current losses at the TFB/ZnO:PEIE interface can hardly limit the PeLEDs performance at the low voltage working condition of FAPbI₃ PeLEDs. However, the current density of the ZnO/TFB only device quickly rises with the increasing voltage due to decreased interfacial potential barrier (Fig. R8) and reaches the same order of magnitude as the perovskite device at larger bias. This increased leakage current might be an important limitation of the EL performance and stability of the low-coverage PeLEDs under intense electronic excitation (*Nat. Photon.* **13**, 418-424 (2019), *Nat. Commun.* **12**, 5081 (2021)).

As such, we believe that the results presented in Supplementary Fig. 12 and 13 do not contradict our argument that devices with low surface coverage are prone to show current shunting issues under high voltage. In addition, the high performance observed in low-coverage films at low voltage is likely due to the influence of additional factors, such as different defect densities, that are inevitably introduced during the control of film morphology.

Supplementary Figure 11. Current density vs voltage curves of the FA-2.4 device and ITO/ZnO/PEIE/TFB/MoO₃/Au device.

Fig. R8 Schematic band energy diagram of ZnO/PEIE/TFB under different biases.

2, This article mainly explains the role of TFA⁻ anions, but does not mention the role of Cs⁺ cations. If Cs⁺ ion does not affect the device performance, why not use FATFA, but CsTFA, which is different from the main cation of perovskite?

Reply: We appreciate the referee for raising this critical concern, which made us realize that the effect of Cs⁺ should be better highlighted. We actually prepared reference devices with CsI (FC devices, Supplementary Fig. 1) instead of CsTFA to distinguish the effects from TFA anions and Cs cations. We did observe distinct increases in radiance reaching 1096.3 W Sr⁻¹ m⁻² by introducing Cs⁺ alone, in contrast to the C-devices (~ 700 W Sr⁻¹ m⁻²). Such an improvement is believed to be correlated with a change of Goldschmidt tolerance factor that improves the tolerance of perovskite regarding to heat and/or large currents (Nat. Commun. 10, 2818 (2019), Adv. Mater. 32, 1907786 (2020), Joule, 8, 1176-1190 (2024)).

Despite clear improvements with Cs⁺, we believe that this effect is still mild if compared with that of TFA⁻ anions. The optimized FCT-devices show a notably higher peak EQE of 21.4% compared with that of F- and FC-ones (~18%). With increasing current density, the EQE values of FCT-devices remain above 20% until reaching a current density as high as ~2,270 mA cm⁻² (Fig. 1c). The devices exhibit minimal current-efficiency roll-off with EQE reductions of approximately 5% at a high radiance of around 2,000 W sr⁻¹ m⁻², a stark contrast to the significant efficiency drop in F- and FC- devices (Fig. 1d). The negligible current-efficiency

roll-off gives rise to a high peak radiance of $\sim 2,409 \text{ W sr}^{-1} \text{ m}^{-2}$ (Fig. 1e). While the addition of CsI provides some benefits, we believe that the impressive performance resulting from CsTFA is primarily due to the TFA⁻ anions rather than the Cs⁺ cations (Supplementary Fig. 1).

We now have provided more discussions about the effect of Cs⁺ in improving the radiance in Page 5, line 100-102.

Supplementary Figure 1. Performance of FC-PeLEDs. a, EQE plotted against current density (EQE-J) for FC-devices. **b,** Dependence of current density and radiance on the voltage (J/R-V) of FC-devices. Here, FC denotes CsI based devices.

3, Is the author's interpretation of Supplementary Fig. 15 and Fig. 4a inverted? XPS is a surface test that cannot indicate the presence of TFA⁻ inside the perovskite.

Reply: We apologize for the misunderstanding caused by our unclear expression. We conducted both X-ray photoelectron spectroscopy (XPS) and time-of-flight secondary ion mass spectrometry (ToF-SIMS) measurements, aiming to verify the presence of TFA⁻ in the perovskite films and investigate their distribution perpendicular to the substrate. Firstly, XPS measurements proved the presence of TFA⁻ on the surface of the perovskite films. Different from the C=O peaks of F-Films, the core-level spectra of C 1s and F 1s of FCT-Films, as depicted in Fig. 4a, clearly reveal the distinct features corresponding to $-\text{O}-\text{C}=\text{O}$ and $-\text{CF}_3$ functional groups. Furthermore, ToF-SIMS measurements show that the TFA⁻ were evenly distributed throughout the perovskite films (Supplementary Fig. 15). These results not only confirm the incorporation of TFA⁻ into the perovskite films but also help to establish the connection to the modified recombination dynamics and film morphology.

To avoid the misunderstanding for the readers, we have polished the corresponding

description in the revised manuscript – page 11, Line 229 – 237.

4, In the XPS test, the author proves the existence of TFA⁻ by the appearance of –O–C=O characteristic peak, but in the components of perovskite, 5AVAI also contains –O–C=O, thus the author needs to avoid the influence of 5AVAI to show that this characteristic peak comes from TFA⁻.

Reply: We are grateful to the referee for this valuable comments. In order to address this concern, we have provided additional FT-IR details to verify the presence of TFA groups in the perovskite films. As shown in the modified Fig. 4a (right panel for C 1s core line spectra), the signals at 292.5 eV and 288.7 eV can be attributed to the –CF₃ group. Additionally, we have provided the F 1s XPS core line spectra, where the peak at 688.8 eV is assigned to the –CF₃ group.

Following the reviewer’s suggestion, we have added more descriptions in the revised manuscript – page 11, Line 232 – 234.

Fig. 4a, Core level spectra of F 1s and C 1s obtained from high-resolution XPS.

Reviewer #3 (Remarks to the Author):

Li et al. have demonstrated perovskite LEDs that maintain a high external quantum efficiency (EQE) of over 20% at current densities exceeding 2 A/cm². This work represents a significant milestone in the perovskite LED field and highlights the potential of perovskite LEDs for practical use. The reviewer strongly recommends the publication of this work without any

reservations.

Reply: We are grateful to the Referee for the positive evaluation of our manuscript.

Reviewer #1 (Remarks to the Author):

In the revised manuscript, the authors have supplemented additional experiments and discussion to improve the manuscript. They addressed most questions raised in the first round of revision, and detailed explanations were included in the response letter. In particular, the response and explanation to the mechanism of balanced charge injection were thoroughly discussed. However, the mechanism on retarded Auger recombination should be further strengthened. Therefore, the authors still need to address the following comments.

1. Although the changes on electron density are highlighted in Fig. 4b, the analysis of charge distribution is still qualitative and unclear. It is also confusing that all charge around I ions are disappeared after TFA anions adsorbed on the perovskite surface. From the results shown in Fig. 4b, TFA can exhibit a large interaction range over 10 Å, is this reasonable? Please provide more convincing evidence and a more detailed analysis of the effective interaction range of TFA.

Reply: We thank the reviewer for the comment. In Fig. 4b, compared to those with clean surfaces, the electron cloud density of targeted cases redistributed and somehow polarized toward TFA due to its strong electron-withdrawing ability. It hence leads to a reduction of electron cloud density in the lattice. It looks that all charge around I ions are disappeared because electron cloud density around I ions is very small and became difficult to distinguish with the naked eye at the same scale bar with other images.

For density functional theory (DFT) calculations, we use the most stable Pb–I terminated perovskite surface and investigate the adsorption geometries for TFA molecule. The hydroxyl group adsorbs on Pb and to fill up the iodine vacancy defect, and the alkyl chain extends in the direction normal to the perovskite crystals with the trifluoromethyl pointing out of plane. In order to avoid confusion, we replace the images in Fig. 4b.

Fig. 4b. Lattice structures and corresponding calculated electron cloud density for FAPbI_3 , $\text{FA}_{1-x}\text{Cs}_x\text{PbI}_3$, and $\text{FA}_{1-x}\text{Cs}_x\text{PbI}_3$ with TFA^- adsorption, respectively. The isosurface is $0.1 \text{ eV}/\text{\AA}^3$.

2. Also in Fig. 4b, it is recommended to provide direct evidences, such as on Bader charge analysis or electron localization function (ELF) to show the quantitative changes of charge quantity. Besides, the legend is missing.

Reply: We are grateful to the Referee for this valuable comment. Electron charge density is the amount of charge of electrons per unit volume, which is used to describes the electrons distribution of materials. The “legend” is not applicable for this calculation. The selected surface of $0.1 \text{ eV}/\text{\AA}^3$ is used as “the isosurface” for electron charge density calculation, which is added into this manuscript.

Electron localization function (ELF) is used to describe the probability of finding an electron with the same spin near a reference position in multi electron systems, which represents the spatial localization degree of measurable reference electrons. The value of ELF ranges from 0 to 1. Here “ $\text{ELF}=1$ ” suggests complete electron localization, while “ $\text{ELF}=0$ ” indicates complete delocalization of electrons. The intermediate value of “ $\text{ELF}=0.5$ ” exhibits the formation of the electron pair distribution similar to the electron gas. Therefore, ELF can be used to determine the type of atom bonding. Following the suggestion from the reviewer, we provide the electron localization function (ELF) and the Bader charge analysis (Supplementary Fig. 16) of the samples to provide information about the electron localization and the bond strength.

From Supplementary Fig. 16, we find that the incorporation of TFA obviously changed the electron distribution of lattices. The electron distribution suggests that bonds in lattices are primarily ionic. The Bader charge analysis indicates that the atoms of the FAPbI_3 loss electrons in the presence of TFA. The dangling F atoms of the TFA have a large electronegativity and the strong ionic bonds are formed with Pb atoms, resulting in the reconstructing in FAPbI_3 lattices. We have now added the corresponding description in this manuscript (Page 11-12, Line 248-251).

Supplementary Figure 16. Electron localization function (ELF) and effective Bader charge for FAPbI_3 , $\text{FA}_{1-x}\text{Cs}_x\text{PbI}_3$, and $\text{FA}_{1-x}\text{Cs}_x\text{PbI}_3$ with TFA^- adsorption, respectively.

3. The author should provide the specific values of the exciton binding energies.

Reply: We thank the referee for raising this question which makes us realize that our data could be better presented. We investigated (Supplementary Fig. 15) the PL intensity as a function of temperature from 120 K to 300 K. Our analysis on temperature-dependent PL enabled us to determine the exciton binding energies, which are approximately ~ 44 meV for F-films and about ~ 28 meV for FCT-films, respectively. The decrease in exciton binding energy often leads to a retarded Auger process due to the changes in Coulomb interactions within the materials. As such, the variations in the exciton binding energy of the perovskites agree well with our calculation results regarding the reduction of electron cloud density on perovskites. We added the corresponding description in this manuscript (Page 11, Line 245-248).

Supplementary Fig. 13 PL emission intensity as a function of temperature for F and FCT films.